# Multidisciplinary Consulting Team for Complicated Cases of Neurodevelopmental and Neurobehavioral Disorders: Assessing the Opportunities and Challenges of Integrating Pharmacogenomics into a Team Setting

**DOI:** 10.3390/jpm12040599

**Published:** 2022-04-08

**Authors:** Pritmohinder S. Gill, Amanda L. Elchynski, Patricia A. Porter-Gill, Bradley G. Goodson, Mary Ann Scott, Damon Lipinski, Amy Seay, Christina Kehn, Tonya Balmakund, G. Bradley Schaefer

**Affiliations:** 1Department of Pediatrics, University of Arkansas for Medical Sciences, Little Rock, AR 72202, USA; balmakundtonyam@uams.edu (T.B.); schaefergb@uams.edu (G.B.S.); 2Arkansas Children’s Research Institute, Little Rock, AR 72202, USA; portergillpa@archildrens.org; 3Arkansas Children’s Hospital, 1 Children’s Way, Little Rock, AR 72202, USA; elchynskia@archildrens.org; 4Schmieding Developmental Center, Springdale, AR 72762, USA; bggoodson@uams.edu (B.G.G.); scottmaryann@uams.edu (M.A.S.); dlipinski@uams.edu (D.L.); adseay@uams.edu (A.S.); kehnchristina@uams.edu (C.K.); 5Arkansas Children’s Hospital Northwest, Springdale, AR 72762, USA; 6University of Arkansas for Medical Sciences Northwest, Fayetteville, AR 72701, USA

**Keywords:** neurodevelopmental disorders, autism spectrum disorder, ADHD, pharmacogenomics, CYP2D6, CYP2C19, phenoconversion

## Abstract

Neurodevelopmental disorders have steadily increased in incidence in the United States. Over the past decade, there have been significant changes in clinical diagnoses and treatments some of which are due to the increasing adoption of pharmacogenomics (PGx) by clinicians. In this pilot study, a multidisciplinary team at the Arkansas Children’s Hospital North West consulted on 27 patients referred for difficult-to-manage neurodevelopmental and/or neurobehavioral disorders. The 27 patients were evaluated by the team using records review, team discussion, and pharmacogenetic testing. OneOme RightMed^®^ (Minneapolis, MN, USA) and the Arkansas Children’s Hospital comprehensive PGx test were used for drug prescribing guidance. Of the 27 patients’ predicted phenotypes, the normal metabolizer was 11 (40.8%) for CYP2C19 and 16 (59.3%) for CYP2D6. For the neurodevelopmental disorders, the most common comorbid conditions included attention-deficit hyperactivity disorder (66.7%), anxiety disorder (59.3%), and autism (40.7%). Following the team assessment and PGx testing, 66.7% of the patients had actionable medication recommendations. This included continuing current therapy, suggesting an appropriate alternative medication, starting a new therapy, or adding adjunct therapy (based on their current medication use). Moreover, 25.9% of patients phenoconverted to a CYP2D6 poor metabolizer. This retrospective chart review pilot study highlights the value of a multidisciplinary treatment approach to deliver precision healthcare by improving physician clinical decisions and potentially impacting patient outcomes. It also shows the feasibility to implement PGx testing in neurodevelopmental/neurobehavioral disorders.

## 1. Introduction

Neurodevelopmental disorders (NDDs) include attention-deficit hyperactivity disorder (ADHD), autism, learning disabilities, mental disabilities, addiction, suicide, depression, conduct disorders, migraine, cerebral palsy, and epilepsy [1,2]. NDDs are characterized by impairments in cognition, communication, behavior, and/or motor skills [3] resulting from abnormal brain development [4], impacting critical signaling pathways and synaptic plasticity [5,6,7]. NDDs are highly heritable and frequently co-occur in individuals, with complex genetic underpinnings [8,9]. From publicly available databases, Leblond and colleagues [10] extracted 1586 high confidence NDD genes, which are expressed at very early stages of fetal brain development and enriched in several biological pathways such as chromosome organization, cell cycle, metabolism, and synaptic function. NDDs are serious health disabilities and last throughout an individual’s lifespan. Almost 17% of children aged 3 through 17 years have one or more developmental disabilities [1] and the estimate for Arkansas children is 27% [11]. The complexity of NDD and the presence of more than one developmental disability increases the difficulty of selecting appropriate pharmacological treatment.

Pharmacogenomics (PGx) is an emerging field of precision medicine, which has great potential for pediatric psychiatry for drug selection and dosing [12,13]. Recent advances in PGx testing provide genuine opportunities in selecting psychotropic medications [14,15], and studies have found that psychiatric disorders can be treated with improved treatment outcomes using genotype-guided therapy [16,17,18]. PGx can predict increased risks of adverse events to psychotropic medications (see Appendix A) and, therefore, prevent therapeutic failure. It is important for patients with NDDs to find a medication that fits each individual and not delay optimal therapy with multiple medication trials. PGx can be a valuable tool to help select treatments to help minimize adverse drug reactions (ADRs) and maximize therapeutic outcomes [17].

Many institutions have developed and implemented multidisciplinary models [19,20,21] with the integration of PGx results into electronic health records along with clinical decision support [12,19,20]. Multidisciplinary management of clinical cases truly lacks full-scale implementation of PGx in pediatric psychiatric clinics. The purpose of this retrospective chart review study was to describe the role of multidisciplinary team consultation aided by PGx testing, especially in “difficult to treat children” with complex neurodevelopmental and/or neurobehavioral disorders (NDD/NBD).

## 2. Materials and Methods

### 2.1. Ethics and Study Design

In July of 2018, the Pediatric Precision Medicine (PPM) clinical service at Arkansas Children’s Northwest (ACNW) was initiated by Dr. Schaefer. The Northwest Arkansas Pediatric Personalized Consulting Service (NWA-PPCS) consists of child psychiatrists, geneticists, neuropsychologists, psychologists, and social workers. The NWA-PPCS multidisciplinary team assessed children with complex (severe) neurodevelopmental, neurobehavioral, and neuropsychiatric disorders. The study was reviewed by the Institutional Review Board (IRB) at the University of Arkansas Medical Sciences (IRB number 262872). The study was determined to be a retrospective chart review only and no identifying patient information would be used in the results. Therefore, a waiver of the Health Insurance Portability and Accountability Act (HIPAA) was granted.

### 2.2. Patient Inclusion/Exclusion Criteria for PGx Referral

For the retrospective chart review study, the patient inclusion criteria for PGx referral included patients less than 21 years old with difficult to control NDDs/NBDs diagnosed per DSM-5 criteria [2], and supporting genetic tests with multiple medication failures or deteriorating health with excessive side effects from medication trials. The patients were referred to the multidisciplinary team as a quaternary medical referral—having already been seen by specialists without success in their treatment from tertiary pediatric providers in neurology, psychiatry, developmental pediatrics, or psychology. Exclusion criteria were simply patients who did not meet inclusion criteria.

### 2.3. Patient Consent

Following NWA-PPCS multidisciplinary team discussions and recommendations, patients were deemed appropriate for PGx testing, and typical genetic testing consent for the PGx testing was obtained from the parent/guardian. Medical consent was a two-step process. First, verbal consent was obtained after describing the purpose and potential benefit of PGx testing for improving the medication therapy. Second, the parent/guardian completed the consent form for PGx testing.

### 2.4. DNA Sample Collection and PGx Testing

Subjects had DNA isolated from blood obtained via standard venipuncture at ACNW in Springdale, Arkansas, USA. Samples were either sent to OneOme, LLC (Minneapolis, MN, USA) for the RightMed^®^ test (https://oneome.com/rightmed-test/; assessed 20 January 2022) or Arkansas Children’s Hospital (ACH) Pathology for the PGx test [12]. The patient ID, date of birth, sex, ethnic background, and medications list were submitted to the OneOme portal or the ACH portal (Figure 1). OneOme analyzes patient DNA on an IntelliQube qPCR platform (Douglas Scientific, Alexandria, MN, USA), which employs 111 TaqMan single-nucleotide polymorphisms (SNPs) genotyping and copy number variation assays for 27 genes (https://oneome.com/rightmed-test/; assessed 20 January 2022). ACH-PGx uses an OpenArray^®^ (ThermoFisher Scientific, Carlsbad, CA, USA) plate using the QuantStudio™12K Flex OpenArray AccuFill System and assays 174 TaqMan^®^ SNPs (ThermoFisher Scientific, Carlsbad, CA, USA) targeting 23 genes [12]. The OneOme and ACH-PGx laboratories are accredited by the College of American Pathologists and certified by the Clinical Laboratory Improvement Amendments (CLIA).

### 2.5. Data Reporting

The genetically polymorphic cytochrome P450 2D6 and 2C19 isozymes (*CYP2D6* and *CYP2C19*) are responsible for the metabolism of numerous psychotropic medications (Appendix A). Actionable variants in *CYP2C19* and *CYP2D6* genes impact medication response and metabolism [22,23]. Table 1 shows *CYP2C19* and *CYP2D6* alleles/variants interrogated for psychotropic medication guidance in the current pilot study. Appendix A shows the respective *CYP2C19* and *CYP2D6* star alleles, and rsIDs; Appendix A shows the CYP2D6 copy number variation assay on the ACH-PGx panel. The wet laboratory results of the RightMed^®^ test and the ACH PGx test were analyzed by a proprietary haplotyping algorithm to convert patient genotypes into diplotype calls. Phenotype assignments were based on a genotype–phenotype translation from the Clinical Pharmacogenetics Implementation Consortium (CPIC). The clinical report with full results showing drug–gene interactions was returned to the host institute (Figure 1).

### 2.6. Clinical Recommendations

The PGx test ordering member subsequently reviewed the results with the NWA-PPCS team members and carefully assessed the patient drug metabolizer status from the PGx report. The drugs with actionable PGx results were discussed with supportive evidence along with the patient’s current medications and therapies, to create a patient-specific comprehensive treatment plan. The patient was said to have an actionable phenotype if their phenotype had psychotropic guidance per the CPIC, Food and Drug Administration (FDA), or Dutch Pharmacogenetics Working Group (DPWG) [24,25,26], for example, CYP2D6 poor metabolizer and aripiprazole. PGx-assisted medication changes were subsequently placed into categories; this included continuing current therapy, suggesting an appropriate alternative medication, starting a new therapy, or adding adjunct therapy. Medications, after PGx testing results were returned, were analyzed for CYP2D6 phenoconversion and actionable phenotypes to see the effect of concomitant medications. As phenoconversion analysis was not part of the clinical protocol and PGx reports, it was assessed after the conclusion of the project to determine the frequency of the CYP2D6 phenoconversion using the publicly available CYP2D6 inhibitor calculator [27].

## 3. Results

### 3.1. Sample Demographics

In total, 27 patients were seen by NWA-PPCS, with a mean age of 11 ± 4 years, and 51.9% were males. Moreover, 85.2% of patients were not Hispanic, and 88.9% were white (Table 2).

### 3.2. Pharmacogenomic (PGx) Results

#### 3.2.1. CYP2C19 and CYP2D6 Phenotype Metabolizer Status

The NWA-PPCS team reviewed the 27 PGx clinical reports and the pharmacogenetic profiles were compared with past and current medication(s). For the psychotropic medications, the focus was on two genes: CYP2C19 and CYP2D6 (Table 1 and Appendix A). Team members discussed the clinical validity of the PGx variants in order to make clinical decisions for patient care and treatment. For example, CYP2C19 analysis showed 11 (40.8%) patients were assigned CYP2C19 normal metabolizer (NM) status (Table 3), and only 7 (25.9%) patients were assigned CYP2C19 intermediate metabolizer (IM) status. CYP2C19 poor metabolizer (PM), rapid metabolizer (RM), and ultrarapid metabolizer (UM) were 3.7%, 25.9%, and 3.7%, respectively. Similarly, for CYP2D6, a total of 16 (59.34%) patients were assigned NM status and 11 (40.7%) were assigned IM status (Table 3). Overall, this pilot study shows 70.4% of patients had at least one clinically actionable phenotype based upon genotype.

#### 3.2.2. Pharmacogenomics Actionability and Phenoconversion

When accounting for CYP2D6 inhibitors, four NMs and three IMs were on a strong CYP2D6 inhibitor (i.e., bupropion, paroxetine, and fluoxetine), and all phenoconverted to a CYP2D6 PMs (25.9%) (Table 4). Of the seven patients who phenoconverted, two were on an additional CYP2D6 psychotropic medication. More than half the patients had an actionable phenotype for CYP2C19 selective serotonin reuptake inhibitors (SSRIs) (59.3%), no patients had an actionable phenotype for CYP2D6 SSRIs and antipsychotics, and 40.7% had an actionable phenotype for atomoxetine. When accounting for CYP2D6 phenoconversion, 25.9% had an actionable phenotype for CYP2D6 SSRIs and antipsychotics, and 55.6% had an actionable phenotype for atomoxetine. Overall, 20 patients had at least one actionable phenotype when taking phenoconversion into consideration.

### 3.3. NDD/NBD Diagnosis

The most common comorbid conditions included ADHD (66.7%), anxiety disorder (59.3%), and ASD (40.7%) (Table 5). Nearly half of the patients had either ASD, intellectual disability, or both. Twenty six percent of the patients had a unifying ‘encephalopathy’ diagnosis.

### 3.4. PGx Guiding Therapy

Some examples of patients’ past diagnoses, genetic test results, and medications before the PGx referral are shown in Table 6. The results of CYP2C19 and CYP2D6 metabolizer status from PGx testing helped guide the NWA-PPCS multidisciplinary team for treatment selection and dosing recommendations. For example, patient 28, who was referred to the NWA-PPCS team, had diagnoses of ASD, ADHD, and anxiety. This patient’s chromosomal microarray analysis was normal and the patient was on risperidone and dexmethylphenidate. The results of PGx show this patient’s metabolizer status was IM for CYP2C19 and NM for CYP2D6 (Table 6). After reviewing the actionable nature of the variants, the team was able to confirm the medication therapy. Patient 30, before PGx testing, was on bupropion with little benefit and PGx results showed a CYP2C19 IM and a clinical phenotype of PM for CYP2D6. The team was able to use the results to select an appropriate adjunct therapy agent to help control the patient’s symptoms. The review of PGx results with potentially actionable variants was able to help guide medication therapy 66.7% of the time to either confirm current therapy (11.1%), select an appropriate alternative (18.5%), new medication start (22.2%), or add adjunct therapy (14.8%) (Table 7).

## 4. Discussion

Notably, a higher proportion of developmental disabilities occur in Arkansas compared with the rest of the USA [1,11] and cause tremendous social and economic burden. The NWA-PPCS team was designed to assess the feasibility and yield of the evaluations of comprehensive neurogenetic and behavioral/psychiatric patients. The intent was to determine if we could customize or optimize medical treatment and supportive therapies and interventions in the most difficult to treat children with NDD/NBD. This team is unique, being one of only a very few clinics in the country that formally incorporates genomic-directed medicine into its work process. It is likely the only one with a dedicated focus on neurodevelopmental and neurobehavioral disorders. The NWA-PPCS small sample study describes the real-world experience with an opportunity to employ PGx testing in a quaternary care model to improve clinical decisions by selecting medication therapy in NDD/NBD patients with medical complexity and history of medication failure or multiple adverse reactions to past medications (Table 6).

NDDs are clinically and genetically heterogenous with a number of gene mutations as well as chromosomal structural variants, which disrupts synaptic plasticity [5,6,7]. Medical comorbidities are common in children with ASD [28], and recognizing those are of paramount importance to have proper therapeutic intervention in difficult to treat psychiatric cases. ADHD and anxiety disorders occurred in 66.7% and 59.3% of patients, respectively, whereas ASD only occurred in 40.7% of patients (Table 5). The most common coexisting psychiatric disorders in subjects with ASD include ADHD [29] and share about 50–72% of their genetic factors [30]. Both the co-occurring disorders ASD and ADHD show high prevalence and between 30–50% of individuals with ASD manifest ADHD symptoms [31]. Evidence of significant comorbidity between anxiety disorders and ASD was found in a meta-analysis [32] and showed 40% of the cases with ASD are diagnosed with at least one anxiety disorder. Our pilot study also showed that 33.3% of patients had an intellectual disability (Table 5). Recent studies show about one percent of the general population with an intellectual disability, and about 10% of those individuals have a diagnosis of ASD or autistic traits [33,34].

Phenoconversion is rightly referred to as the Achilles’ heel [35] and it converts genotypic CYP2D6 NM to phenotypic IM or PM, or genotypic CYP2D6 IM to phenotypic PM due to the concomitant use of a CYP2D6 inhibitor. The conversion to a different phenotype can impact how a patient will respond to a medication and potentially affect the therapeutic outcome [36]. The patients in this study were genotyped as NM and IM for CYP2D6 (Table 3), with 25.9% of patients phenoconverting to a CYP2D6 PM (Table 4). Antidepressants causing phenoconversion in our patient population were paroxetine, fluoxetine [37], and bupropion [38]. These three agents are strong CYP2D6 inhibitors as defined by the Food and Drug Administration (FDA) [39] that caused the phenoconversion of the patients to CYP2D6 PMs. Two of the patients in the current study who underwent phenoconversion were on other medications metabolized via CYP2D6 and potentially may have had undesired side effects.

Almost two-thirds of patients in this retrospective study carry one actionable variant impacting drug response. When utilizing PGx for medication therapy management, the NWA-PPCS multidisciplinary team demonstrate improved clinical decisions by following the use of PGx actionable guidelines [24,25,26] for confirming treatment, choosing an alternative medication, new medication start, or adding an adjunct psychotropic medication in 66.7% of patients (Table 6). When comparing our results with patients with major depressive disorder (MDD) and anxiety, a lower agreement was found in selecting medications congruent with PGx. Dagar and colleagues reported that 43.7% of patients had new medications added and 32.7% replaced based on PGx, which led to improved clinical outcomes [16]. The NWA-PPCS multidisciplinary team recommended a change in medication (i.e., new medication start or adjunct) in 37% and alternative therapy in 18.5%. However, Brown and colleagues. [40] reported that after PGx testing, 81% of patients were on therapy congruent with the patient’s PGx results. A potential reason for our rates of using PGx being lower is that SSRIs are not necessarily first-line therapy for patients with NDD. Preferred first-line agents include antipsychotics, alpha two agonists, and stimulants [41,42] compared with MDD and anxiety, for which SSRIs are recommended first-line [43,44]. Roscizewski and colleagues reported that 53.8% of patients experienced a change in medication based on PGx, which is similar to our result of 55.6% [45]. These studies show the usefulness of PGx guidance in patients who are beginning new therapy or who are not responding to therapy for NDD/NBD.

The retrospective chart review study from the NWA-PPCS multidisciplinary team has several limitations. Firstly, the study’s primary limitation is that it was a retrospective chart review of patients who underwent PGx testing. This study design limits outcome data that would be available if this study was designed as a prospective cohort or a randomized controlled trial. However, this population has not been studied previously, and our study can help support future studies on this difficult-to-treat population. A second limitation is the relatively small size of NDDs/NBDs patients in our study, and therefore, limited in its ability to demonstrate results. Our small sample size can be explained by the study design being a retrospective chart review and having only one clinic adopting PGx-testing in patients with NDDs/NBDs. Future larger prospective/clinical trial studies could help further demonstrate the need to test patients with NDDs/NBDs. The third limitation of having a control group is beyond the scope of this feasibility study. A control group would have required longitudinal assessment outcomes for both groups, either by collecting pharmacokinetic or pharmacodynamic data. Our study is also limited in not having pharmacokinetic data on the medications prescribed that have pharmacogenomic guidance and not having pharmacodynamic outcomes in the patients. This limitation is due to the trial design being a retrospective chart review. Considering these parameters, further studies will need to be conducted to help demonstrate if PGx-testing in this population has benefits. Finally, as previously mentioned, phenoconversion was not part of the PGx clinical report, and conversion to a different phenotype can change the therapeutic outcome. It is becoming clear that implementing PGx analysis algorithms should incorporate drug induced phenoconversion with rigorous validations. Given these limitations, there is a great promise for PGx testing to provide drug and dosing guidance for difficult to treat patients with NDDs/NBDs.

## 5. Conclusions

Without a doubt, the field of pharmacogenomics has great potential for enhancing patient care in any field of medicine. This would be particularly true in the realm of persons with NDDs/NBDS. In this retrospective chart review study, the NWA-PPCS multidisciplinary team found a high level of effectiveness for interdisciplinary consulting augmented with PGx testing in guiding medication therapy in difficult to treat psychiatric conditions, which facilitated understanding of comorbid associations and choice for individualized treatment options.

The pharmacogenetic profiles helped influence the prescribing practices moving forward with the clinical management of patients with NDDs/NBDs. The NWA-PPCS multidisciplinary clinic was able to successfully adopt PGx into clinical care to deliver precision medicine at the Arkansas Children’s Hospital North West. Challenges encountered that should be considered by other institutions implementing PGx are uploading the data discretely and developing clinical decision support to have PGx guidance at the time of prescribing.

Further studies are required to verify the benefit of multidisciplinary consultation and PGx guidance in treatment decisions for NDDs/NBDs.

## Figures and Tables

**Figure 1 jpm-12-00599-f001:**
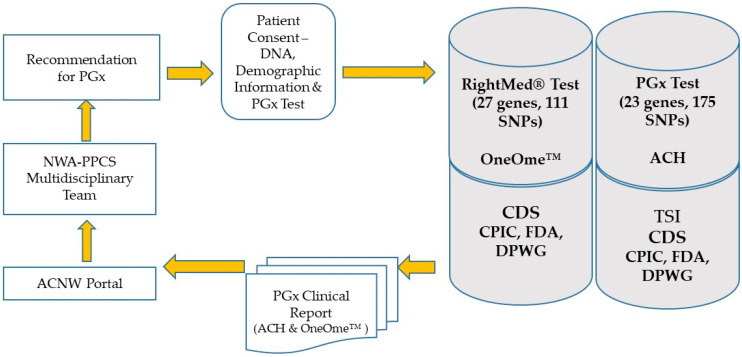
Study design for ACNW pilot project. All patients were under 21 years old, had a PGx test as part of their clinical care, and had a consultation with the NWA-PPCS. PGx = pharmacogenomics; ACNW = Arkansas Children’s Northwest; ACH= Arkansas Children’s Hospital; TSI = Translational Software, Inc., (Bellevue, WA, USA); CDS = clinical decision support; CPIC = Clinical Pharmacogenetics Implementation Consortium; DPWG = Dutch Pharmacogenetics Working Group; FDA = Food and Drug Administration.

**Table 1 jpm-12-00599-t001:** *CYP2C19* and *CYP2D6* star alleles (*) on the RightMed^®^ Test and the ACH PGx panel.

Gene	RightMed^®^ Alleles	ACH PGx Alleles
*CYP2C19*	* 2, * 3, * 4, * 4B, * 10, * 17	* 2, * 3, * 4A, * 4B, * 5, * 6, * 7, * 8, * 9, * 10, * 11, * 17, * 35
*CYP2D6*	* 2, * 2A, * 3, * 4, * 4J, * 4K, * 4M, * 4N, * 5, * 6, * 6C, * 7, * 8, * 9, * 10, * 11, * 12,* 13, * 14, * 15, * 17, * 18, * 19, * 29, * 31, * 34, * 35, * 36, * 39, * 41, * 42, * 59, * 61, * 63, * 64, * 65, * 68, * 69, * 70, * 91, * 109, * 114, CNVs	* 4, * 6, * 7, * 8, * 9, * 10, * 11, * 12, * 14A, * 14B, * 15, * 17, * 18, * 19, * 21, * 29, * 30, * 31, * 33, * 35, * 36, * 38, * 40, * 41, * 42, * 43, * 44, * 45, * 46, * 47, * 49, * 51, * 53, * 54, * 56A, * 56B, * 58, * 62, * 70, * 84, * 100, * 101, CNVs

CNVs = Copy Number Variations Assay.

**Table 2 jpm-12-00599-t002:** Patient Demographics.

Patient Characteristics	*n* = 27
Age, years (AVG ± SD)	11 ± 4
Sex (%)	
Males	14 (51.9)
Race (%)	
White	24 (88.9)
Other	3 (11.1)
Ethnicity (%)	
Not Hispanic	23 (85.2)

AVG = average, SD = standard deviation.

**Table 3 jpm-12-00599-t003:** Assigned likely patient CYP2C19 and CYP2D6 phenotypes.

Gene (Phenotype)	*n* (%)
CYP2C19 Phenotype	
PM	1 (3.7)
IM	7 (25.9)
NM	11 (40.8)
RM	7 (25.9)
UM	1 (3.7)
CYP2D6 Phenotype	
PM	0 (0)
IM	11 (40.7)
NM	16 (59.3)
UM	0 (0)

PM = poor metabolizer; IM = intermediate metabolizer; NM = normal metabolizer; RM = rapid metabolizer; UM = ultrarapid metabolizer.

**Table 4 jpm-12-00599-t004:** CYP2D6 phenoconversion.

Gene (Phenotype)	*n* (%)
CYP2D6 Phenoconversion	7 (25.9)
*Clinical CYP2D6 Phenotype	
PM	7 (25.9)
IM	8 (29.6)
NM	12 (44.4)
UM	0 (0)

PM = poor metabolizer; IM = intermediate metabolizer; NM = normal metabolizer; UM = ultrarapid metabolizer; *clinical CYP2D6 phenotype is the phenotype for CYP2D6 based on genotype and concomitant medications generated from the publicly available CYP2D6 calculator [25].

**Table 5 jpm-12-00599-t005:** Most common diagnosis for NDDs/NBDs.

Most Common Diagnosis	*n* (%)
ADHD	18 (66.7)
Anxiety Disorder	16 (59.3)
Autism	11 (40.7)
Intellectual Disability	9 (33.3)
Sleep Difficulties	8 (29.6)
Encephalopathy	7 (25.9)

The diagnosis for autism, ADHD, anxiety disorder, intellectual disability, and sleep difficulties was conducted per DSM-5 criteria. A diagnosis of a static (non-metabolic or inflammatory) encephalopathy was used in the context of patients with multiple inter-related NDDs/NBDs.

**Table 6 jpm-12-00599-t006:** Examples of PGx adoption by the NWA-PPCS multidisciplinary team for guidance of therapy in patients with NDD/NBD.

PatientID	Diagnoses	Diagnostic Genetic Testing Results	Medications Before PGx	CYP2C19 Phenotype	CYP2D6 Phenotype	Medications After PGx	PGx Medication Change Classification
7	Encephalopathy Spastic paraparesis Panic Disorder ADHD Borderline Personality Disorder	*PUM1* related disorder	None	NM	NM	Escitalopram 5 mg, 1 tablet daily Trazodone 50 mg, 1 tablet at bedtime	New Start
12	Middle cerebral artery syndrome Encephalopathy Migraine Disorder Growth Hormone Disorder ADHD Social Anxiety Mild Intellectual Disability	*KAT8* related disorder	Guanfacine ER 4 mg, 1 table at night Venlafaxine ER 150 mg, 1 tablet every morning Trazodone 150 mg, 1 tablet at bedtime	IM	IM	Guanfacine ER 4mg, 1 tablet at bedtime Trazodone 150 mg, 1 tablet at bedtime Duloxetine 30 mg, 1 tablet daily	Alternative Therapy
26	Encephalopathy ASD ADHD Borderline Intellectual Functioning Sleep Difficulties	Exome sequencing non-diagnostic	Methylphenidate 10 mg, 1 tablet two times daily	IM	IM	Methylphenidate 10 mg, 1 tablet two times daily Clonidine 0.1 mg, 1 tablet at bedtime Hydroxyzine 25 mg, 1 tablet two times daily Risperidone 0.5 mg, 1 tablet two times daily	Adjunct Therapy
28	ASD ADHD Anxiety	Chromosome Microarray Analysis Normal	Risperidone 0.5 mg, 2 tablets daily Dexmethylphenidate XR 5 mg, 1 tablet daily	IM	NM	Risperidone 0.5 mg, 2 tablets daily Dexmethylphenidate XR 10mg, 1 tablet daily	Confirmed Therapy
30	ADH Anxiety Cognitive Disorder Executive Function Deficient Learn Disorder involving mathematics	No additional genetic testing	Bupropion 400 mg daily	IM	PM *	Bupropion 400 mg daily Guanfacine XR 3 mg, 3 tablets by mouth at bedtime	Adjunct Therapy

NDD = neurodevelopmental disorder; NBD = neurobehavioral disorder; ADHD = attention-deficit hyperactivity disorder; ASD = autism spectrum disorder; *PUM1* = Pumilio RNA binding family member 1; *KAT8* = lysine acetyltransferase 8; XR or ER = extended release; CYP2D6 and CYP2C19 Phenotype IM = intermediate metabolizer; NM = normal metabolizer; * clinical phenotype based on the patients genotype result and concomitant medication.

**Table 7 jpm-12-00599-t007:** PGx informing the therapy decisions.

Utilized PGx to Guide Therapy	*n* = 18 (66.7)
New medication start	6 (22.2)
Alternative therapy	5 (18.5)
Adjunct therapy	4 (14.8)
Confirmed therapy	3 (11.1)

## Data Availability

On reader’s request, PGx results on CYP2C19 and CYP2D6 can be shared.

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
