# Peer review of "Multidisciplinary Consulting Team for Complicated Cases of Neurodevelopmental and Neurobehavioral Disorders: Assessing the Opportunities and Challenges of Integrating Pharmacogenomics into a Team Setting"

_jpm, 2022, doi:10.3390/jpm12040599_

Round 1

Reviewer 1 Report

In this pilot study, a multidisciplinary team consulted on patients with difficult-to-manage neurodevelopmental and/or neurobehavioral disorders. This pilot study highlights the value of a multidisciplinary treatment approach including PGx testing to deliver precision healthcare by improving physician clinical decisions that potentially impact patient outcomes. NWA-PPCS multidisciplinary clinic was able to successfully integrate PGx into clinical care to deliver precision healthcare for Arkansas Children’s, this fact was so excellent.

Abstract

The typo was found in Abstract: “difficlt-to-manage neurodeveopmental”, “neurobehaviotal”

Please check the spelling of the terms in the manuscript.

Introduction

I think some references are required for the sentence in lines 45-46.

Please check the format for reference (i. e., “guided therapy.[7-9].”, “dosing [10-11].” “outcomes[10].”)

Is hyphen required for “difficult to treat children”?

Line 66: “and / ot”

Line 149; “sinhibitor” may be a typo.

It would be helpful if the summarized table on the relationship between each gene and antipsychotics.

I would like to know whether the combination of genotype for each gene affects the difficulty of pharmacologic medication.

Are there any patients who arrange the drug dosage?

The author should describe the strategy for “PGx informing the therapy decisions” in detail.

Line 170: The author should mention “behavioral therapy” in this NWA-PPCS team.

Lines 176-177: How many patients relief from the AE of previous medication? The AE was associated with the genotype of any genes?

Lines 200-202: In my opinion, the result of “phenoconversion” may be better described in the result section, if the conversion strategy was planned in the study protocol.

To discuss the contribution of PGx intervention to the antipsychotic medication, the author should describe which drugs were more critically affecting the phenoconversion practice in the result section, which may show the priority of PGx testing.

Line 229  integrate? healthcare?

Author Response

Thank you very much for your constructive comments and suggestions. Please find below our responses:

Reviewer 1:

Comments and Suggestions for Authors

In this pilot study, a multidisciplinary team consulted on patients with difficult-to-manage neurodevelopmental and/or neurobehavioral disorders. This pilot study highlights the value of a multidisciplinary treatment approach including PGx testing to deliver precision healthcare by improving physician clinical decisions that potentially impact patient outcomes. NWA-PPCS multidisciplinary clinic was able to successfully integrate PGx into clinical care to deliver precision healthcare for Arkansas Children’s, this fact was so excellent.

  • Abstract

\The typo was found in Abstract: “difficlt-to-manage neurodeveopmental”, “neurobehaviotal”

Please check the spelling of the terms in the manuscript.

Thank you for bringing this error to our attention. It has been corrected on line 24.

We have corrected the neurodevelopmental/neurobehavioral spellings in the manuscript.

  • Introduction

I think some references are required for the sentence in lines 45-46.

Than you for pointing this out, we have added references as suggested by the your (see lines 46-53) and also, please see References: 4-9 (lines 352-371).

  • Please check the format for reference (i. e., “guided therapy.[7-9].”, “dosing [10-11].” “outcomes[10].”)

Thank you for pointing out the reference format. We have corrected this spacing mistake (lines 62 and 65).

  • Is hyphen required for “difficult to treat children”?

Thank you for your suggestion, we have added a hyphen.

  • Line 66: “and / ot” Thank you for bringing this error to our attention. We have corrected it, and is now on line 78

  • Line 149; “sinhibitor” may be a typo. Thank you for bringing this error to our attention. We have corrected it, and it is now on line 197.

  • It would be helpful if the summarized table on the relationship between each gene and antipsychotics.

Thank you for this suggestion. We have created a Table showing drug-gene interaction as Supplementary Table S1 to help describe the relationship between the genes and psychotropic medications. Also, we have added this information in text on lines 66, 142, 177 and 320. (see Supplementary materials as Table S1).

  • I would like to know whether the combination of genotype for each gene affects the difficulty of pharmacologic medication.

Of course, the combination of genotypes affects drug metabolism. Some drug-gene interactions become complex when two genes are involved.

For example, drug amitriptyline uses both CYP2D6 and CYP2C19 phenotypes to make therapeutic recommendations (Hicks et al. 2017 CLINICAL PHARMACOLOGY & THERAPEUTICS | VOLUME 102 NUMBER 1). For example, the CPIC guidelines for amitriptyline dosing show for CYP2C19 Intermediate metabolizer and CYP2D6 ultrarapid metabolizer recommend avoid amitriptyline, and when CYP2C19 and CYP2D6 are both intermediate metabolizers, they recommend consider a 25% reduction in starting amitriptyline dose.

  • Are there any patients who arrange the drug dosage?

Our apologies, we did not understand what was being asked. May you please elaborate?

  • The author should describe the strategy for “PGx informing the therapy decisions” in detail.

Thank you for this comment and suggestion.

We have added Supplementary Table S2, which shows patients’ diagnoses, genetic test results, and treatments before PGx testing. This shows how the NWA-PPCS multidisciplinary team used the CYP2C19 and CYP2D6 metabolizer status to help guide psychotropic medications.

See lines 61, 224, 252, 321.

  • Line 170: The author should mention “behavioral therapy” in this NWA-PPCS team.

As mentioned, who is involved in our NWA-PPCS team, psychologists perform behavioral therapy as part of their standard practice, and performing behavioral therapy is understood.

  • Lines 176-177: How many patients relief from the AE of previous medication? The AE was associated with the genotype of any genes?

Lines 251-252;

We describe these patients as seriously ill in lines 99-104 (see section on inclusion/exclusion criteria). As such, we did not have any genotype information to say which genotype/gene-drug was giving adverse effects.

Patient inclusion criteria for PGx referral included patients less than 21 years old with difficult to control NDDs / NBDs who had multiple medication failures or with deteriorating health with excessive side effects from medication trials.  The patients were referred to the multidisciplinary team as a quaternary medical referral – having already been seen by specialists without success in their treatment from tertiary pediatric providers in neurology, psychiatry, developmental pediatrics or psychology. Exclusion criteria were simply patients who did not meet inclusion criteria.  

  • Lines 200-202: In my opinion, the result of “phenoconversion” may be better described in the result section, if the conversion strategy was planned in the study protocol.

Now line 195:

Phenoconversion was not part of the initial clinical protocol, and the PGx reporting platform also did not include phenoconversion. Analyzing if the patient underwent phenoconversion was done at the end of the study when examining the data to see the effect of concomitant medications and describe the prevalence in our retrospective chart review study

See lines 162-165, 190, 195, 209 and 265

  • To discuss the contribution of PGx intervention to the antipsychotic medication, the author should describe which drugs were more critically affecting the phenoconversion practice in the result section, which may show the priority of PGx testing.

The result section describes these results as subheading Phenoconversion lines- 196-197; 270-275.

For clarity, we have divided it into its own separate Table 4.

Also, in the discussion, we have shown this as lines 220-275 as below:

The patients in this study were genotyped as NM and IM for CYP2D6 (Table 3), with 25.9% of patients phenoconverting to a CYP2D6 PM (Table 4). Antidepressants causing phenoconversion in our patient population were paroxetine, fluoxetine [32], and bupropion [33]. These three agents are strong CYP2D6 inhibitors as defined by the Food and Drug Administration (FDA) (34)

  • Line 229  integrate? healthcare?

We have switched integrate to adopt.

S

Reviewer 2 Report

Dear authors;

Manuscript entitled “Multidisciplinary Consulting Team for Complicated Cases of Neurodevelopmental and Neurobehavioral Disorders: Assessing the Opportunities and Challenges of Integrating Pharmacogenomics into a Team Setting”
I was asked to review the manuscript "Multidisciplinary Consulting Team for Complicated Cases of Neurodevelopmental and Neurobehavioral Disorders: Assessing the Opportunities and Challenges of Integrating Pharmacogenomics into a Team Setting" by Belbase, and colleagues. Thus, I have some suggestions, which may improve the paper:

Major Comments:
-    The authors need to include the gap and purpose of the study with more clearly.
-    The gap was not included in the abstract.
-    Random sampling with probability proportionate to sample size method was applied to select desired samples?

Background
- My concern is that the paper does not review all the available literature. Many important studies about clinical decision-making based on pharmacogenomics, were not been described. Therefore, the substantial amount of information is missing.
- The authors need to include the gap and purpose of the study. In addition, it needs to make clear the gap.

Methods: 
- Demonstrate the calculation of sampling in the manuscript, and indicate the parameters or programs that were entered for this sampling analysis.
- Comparative analysis between the components of the multidisciplinary team during the objective assessment process should be taken into account. This evaluation for the selection of all patients and allocation in analysis groups must be done through ICC. An internal validation analysis of the questionnaire for the target population should be established by Cronbach alpha, suggestions to the research group.
- The evaluative criteria for the allocation of decision-making to drugs during treatment are superficial.
- It would be interesting to use regression models to generate allocation formulas and classification of groups based on pharmacogenomics, a suggestion.
 - A detailed description of the statistics used, as well as the effect and power to associate this decision-making in the clinical treatment.
- Principal component analysis test to select the variables that are best suited for the evaluation in question.
- Assess whether the gene flow of the population is constant, then analyze the HW equilibrium.

Final considerations
The manuscript is interesting and an important paper. However, it lacks some measures and specifications to foresee the objectives presented.

Sincerely.

Author Response

Thank you very much for your constructive comments and suggestions. Please find below our responses:

Manuscript entitled “Multidisciplinary Consulting Team for Complicated Cases of Neurodevelopmental and Neurobehavioral Disorders: Assessing the Opportunities and Challenges of Integrating Pharmacogenomics into a Team Setting”
I was asked to review the manuscript "Multidisciplinary Consulting Team for Complicated Cases of Neurodevelopmental and Neurobehavioral Disorders: Assessing the Opportunities and Challenges of Integrating Pharmacogenomics into a Team Setting" by Belbase, and colleagues. Thus, I have some suggestions, which may improve the paper:

Manuscript entitled “Multidisciplinary Consulting Team for Complicated Cases of Neurodevelopmental and Neurobehavioral Disorders: Assessing the Opportunities and Challenges of Integrating Pharmacogenomics into a Team Setting”
I was asked to review the manuscript "Multidisciplinary Consulting Team for Complicated Cases of Neurodevelopmental and Neurobehavioral Disorders: Assessing the Opportunities and Challenges of Integrating Pharmacogenomics into a Team Setting" by Belbase, and colleagues. Thus, I have some suggestions, which may improve the paper:

Major Comments:
-    The authors need to include the gap and purpose of the study with more clearly.

The study was a retrospective chart review and was not designed to present Gap analysis for clinical understanding, variant analysis, and disparity populations.

Instead, the purpose was per our IRB approval, a chart review only (lines 89-90), where team members have a solid understanding of PGx results with an onboard Pharmacogenomics expert for clinical decisions.

The study included patients who were referred to the multidisciplinary team as a quaternary medical referral – having already been seen by specialists without success in their treatment from tertiary pediatric providers (Lines 100-105)

-    Random sampling with probability proportionate to sample size method was applied to select desired samples?

No, we did not apply this sampling method. All patients who were referred were with deteriorating health, and multiple drug failures were included in this pilot project. In addition, with it being a chart review, we did not actively recruit patients to undergo genetic testing or recruit for a control group.

Background
- My concern is that the paper does not review all the available literature. Many important studies about clinical decision-making based on pharmacogenomics, were not been described. Therefore, the substantial amount of information is missing.

Thank you for your suggestion on including literature on pharmacogenomic decision-making. We have shown references in psychiatric guidance (see references 12-18 and 30-40. A lot of information on clinical decision-making is part of the references of FDA/CPIC/DPWG (references 22-24)

- The authors need to include the gap and purpose of the study. In addition, it needs to make clear the gap.

This chart review focuses on patients who underwent pharmacogenomic testing as part of clinical care and discusses the feasibility of other teams implanting PGx in those who work with patients with NDD/NBD. To help clarify, we have added more information on the purpose of this study. See lines 72-78, 112.

Methods: 
- Demonstrate the calculation of sampling in the manuscript, and indicate the parameters or programs that were entered for this sampling analysis.
- Comparative analysis between the components of the multidisciplinary team during the objective assessment process should be taken into account. This evaluation for the selection of all patients and allocation in analysis groups must be done through ICC.

The study design was a retrospective chart review for patients as defined in Ethics Section 2.1 and  inclusion/exclusion criteria Section 2.2 lines 98-99. Our purpose was not to compare our patients to those who did not undergo genetic testing and compare outcomes. We instead focused on the design of our team and the feasibility/considerations other clinics should consider when implementing PGx. Comparative analysis is beyond the scope of this paper.

An internal validation analysis of the questionnaire for the target population should be established by Cronbach alpha, suggestions to the research group.

- The evaluative criteria for the allocation of decision-making to drugs during treatment are superficial.
- It would be interesting to use regression models to generate allocation formulas and classification of groups based on pharmacogenomics, a suggestion.
 - A detailed description of the statistics used, as well as the effect and power to associate this decision-making in the clinical treatment.
- Principal component analysis test to select the variables that are best suited for the evaluation in question.
- Assess whether the gene flow of the population is constant, then analyze the HW equilibrium.

As described above, much of what is being recommended is beyond the scope and purposse of our study. In addition, with no control group and limited small sample size, we would not be able to complete the recommended statistics dealing with regression analysis, PCA and gene flow. The recommendations provided will be considered as we move forward to look at large data studies.

Final considerations
The manuscript is interesting and an important paper. However, it lacks some measures and specifications to foresee the objectives presented.

Thank you for your comments and suggestions. Will definitely consider your elegant suggestions on sample design and analysis, when we venture on to a larger epidemiological study addressing many concerns you have raised.

Reviewer 3 Report

The authors present the results from a pilot study that try to apply pharmacogenomics (PGx) testing and drugs administered in patients with neurodevelopmental disorders. The work is interesting towards the field of adopting PG testing in clinical decisions but also shows the advantages of multidisciplinary clinical teams for advanced healthcare provision. The results are interesting. However, the data are poorly presented in my opinion and there are some issues with the bioethics part of the study mostly of how the data were obtained. As the authors state in their conclusion "Without a doubt, the field of pharmacogenomics has great potential for enhancing care in any field of medicine". I agree thus, this kind of studies (as this one) should describe in detail the benefits of applying PGx testing in clinical decision medicine towards the personalized/precision healthcare era.

Comments:

1) Introduction needs additional information stating the aim of the study clearer or edited for a better reading flow. For example lines 54-57 fit better after line 59 "....dosing [10-11]. " Moreover, lines 59-64 need additional examples in references. Finally, how the idea presented that these patients would benefit from PG testing and from a multidisciplinary team consultation, especially for the latter no data are presented in introduction that could further advance the goals of the study. Moreover, since this work is about drugs and PGxs, what drugs are usually used in those cases?

2) Line 74. It is confusing the statement the study was determined not to be human subject research. It is human subject research. It involved a diagnostic test after all. More importantly, it was regarding children or adults less than 21 years old.

3) HIPAA stands for what?

4) Who determined the statements in line 81 and under what criteria? The patients had participated in clinical trials and no medication was administered to them? It is confusing.

5) The study is an analysis of diagnostic technique using personal health data. Why was the informed consent not used? How can we determine that the analysis did not involve anonymized data? And moreover, regarding the results. What about bias? This is strange because in line 247 they state that informed consent was obtained. So, what is it? Please provide at least for the editor an anonymized example of the informed consent form that was used in the study. 2.2. Does not state that the data were anonymized.

6) In what terms only CYP2D6 and CYP2C19 were chosen? Or is it the PGx test used fixed only for these two? Because CYP2C9 also (that has PG info) metabolizes psychotropic medications etc. etc.

7) Phenoconversion is describing a drug-drug interaction (see also lines 192-194), so, what was the clinical significance of those DDIs?Why special focus on Aripiprazole? (Apart the fact that it needs dose fixing with CYP2D6 inhibitors). Generally, the drug information is missing. Moreover, phenoconversion refers also to the genotype-phenotype mismatch in enzyme abundance although the administration of inhibitors does not necessary impacts enzyme abundance but enzymatic activity. Please explain what the case is here. If it refers to co-administration of drug inhibitors, then we talk more for a drug interaction study. Otherwise please provide the relative enzyme abundance data from the PGx analysis.

8) 45. CFR 102. Does the abbreviation stand for Common Federal Rule? Which part of the "§46.102 Definitions for purposes of this policy." describes pharmacogenomic testing as a NOT human subject procedure? Figure 1 describes just that. I am not expert in legal details, especially in that level, but I think (apologies if I am wrong), Pharmacogenomic testing falls within "An identifiable biospecimen is a biospecimen for which the identity of the subject is or may readily be ascertained by the investigator or associated with the biospecimen."

9) Is the PGx results enough to change the administration? Are there any clinical guidelines that propose the use of PGx results to modify it? The authors state that this is not a study in human subjects after all so the clinical documentation should be enough for this case. If not and there are not clinical guidelines then this pilot study is a study in human subjects.  

10) 163-165. It is the essential part of the study, but no additional information is given. Which drugs, what interventions what actual clinical decisions were given based on PGx data.

11) Would expect at least a comparison of the clinical guidelines and the data from this study, how the utilization of PGx information improved the clinical decisions. An example, there are differences in doses since the study was referring to children and some adults. We do not see any discussion about those details that could further add in this study. We only see a brief discussion between this study and other studies with other disorders (lines 203-219).

12) Too many abbreviations without description e.g., SSRIs etc.

Author Response

Thank you very much for your suggestions and comments. Please find below our responses:

The authors present the results from a pilot study that try to apply pharmacogenomics (PGx) testing and drugs administered in patients with neurodevelopmental disorders. The work is interesting towards the field of adopting PG testing in clinical decisions but also shows the advantages of multidisciplinary clinical teams for advanced healthcare provision. The results are interesting. However, the data are poorly presented in my opinion and there are some issues with the bioethics part of the study mostly of how the data were obtained. As the authors state in their conclusion "Without a doubt, the field of pharmacogenomics has great potential for enhancing care in any field of medicine". I agree thus, this kind of studies (as this one) should describe in detail the benefits of applying PGx testing in clinical decision medicine towards the personalized/precision healthcare era.

Comments:

  • Introduction needs additional information stating the aim of the study clearer or edited for a better reading flow. For example lines 54-57 fit better after line 59 "....dosing [10-11].” Moreover, lines 59-64 need additional examples in references. Finally, how the idea presented that these patients would benefit from PG testing and from a multidisciplinary team consultation, especially for the latter no data are presented in introduction that could further advance the goals of the study. Moreover, since this work is about drugs and PGxs, what drugs are usually used in those cases?

Thank you for your suggestion. We have made changes and are now  lines 62-66.

Pharmacogenomics (PGx) is an emerging field of precision medicine, which has great potential for pediatric psychiatry for drug selection and dosing [12, 13]. Recent advances in PGx testing can benefit clinicians in selecting psychotropic medications [14,15], and studies have found that psychiatric disorders can be treated with improved treatment outcomes using genotype-guided therapy- [16-18].

2) Line 74. It is confusing the statement the study was determined not to be human subject research. It is human subject research. It involved a diagnostic test after all. More importantly, it was regarding children or adults less than 21 years old.

We apologize for this confusion. Yes, it is a human subject chart review only; our IRB saw no need for a consent. Patients were consented as part of routine clinical care to order a genetic test.

Please see section added 2.3

  • HIPAA stands for what?

Thank you for this suggestion, the description of HIPPA has been added on lines 90-91.

4) Who determined the statements in line 81 and under what criteria? The patients had participated in clinical trials and no medication was administered to them? It is confusing.

 Our multidisciplinary team received quaternary referrals in patients who required additional consultation.

5) The study is an analysis of diagnostic technique using personal health data. Why was the informed consent not used? How can we determine that the analysis did not involve anonymized data? And moreover, regarding the results. What about bias? This is strange because in line 247 they state that informed consent was obtained. So, what is it? Please provide at least for the editor an anonymized example of the informed consent form that was used in the study. 2.2. Does not state that the data were anonymized.

Clarification provided above (comment 2).

6) In what terms only CYP2D6 and CYP2C19 were chosen? Or is it the PGx test used fixed only for these two? Because CYP2C9 also (that has PG info) metabolizes psychotropic medications etc. etc.

See Clinical recommendations section 2.5: In our implementation strategy at Children’s we used CPIC, DPWG, and the FDA for guidance and the language describing this is on lines 151-158. Per these resources, CYP2C9 has not reached actionable level to be used to make medications changes in patient’s prescribed psychotropic medications. This is why CYP2C9 was not considered when making medication changes.

7) Phenoconversion is describing a drug-drug interaction (see also lines 192-194), so, what was the clinical significance of those DDIs?Why special focus on Aripiprazole? (Apart the fact that it needs dose fixing with CYP2D6 inhibitors). Generally, the drug information is missing. Moreover, phenoconversion refers also to the genotype-phenotype mismatch in enzyme abundance although the administration of inhibitors does not necessary impacts enzyme abundance but enzymatic activity. Please explain what the case is here. If it refers to co-administration of drug inhibitors, then we talk more for a drug interaction study. Otherwise please provide the relative enzyme abundance data from the PGx analysis.

The clinical reports do not provide phenoconversion information (see line 161-165) and was not assessed when modifying patient’s psychotropic therapy. As phenoconversion is an important concept, we wanted to assess how many patients underwent phenoconversion and how many of them may have required a change in therapy based on presence of the CYP2D6 inhibitor. Aripiprazole was mentioned to provide an example of an actionable phenotype and medication combination; it was not mentioned to discuss phenoconversion. Discussing the relative enzyme abundance is beyond the scope of this paper as our aim was to discuss using PGx in a multidisciplinary team. Phenoconversion was looked at after the patients were seen to assess how many patients had a genotype to phenotype mismatch caused by the presences of a CYP2D6 inhibitor.

8) 45. CFR 102. Does the abbreviation stand for Common Federal Rule? Which part of the "§46.102 Definitions for purposes of this policy." describes pharmacogenomic testing as a NOT human subject procedure? Figure 1 describes just that. I am not expert in legal details, especially in that level, but I think (apologies if I am wrong), Pharmacogenomic testing falls within "An identifiable biospecimen is a biospecimen for which the identity of the subject is or may readily be ascertained by the investigator or associated with the biospecimen."

Our IRB found our study a chart review only, and no further consenting for us to examine the medical records was required.

Per CRF 46.102 Exempt research, human subjects research that is classified as “exempt,” means that the research qualifies as no risk or minimal risk to subjects and is exempt from most of the requirements of the Federal Policy for the Protection of Human Subjects, but is still considered research requiring an IRB review for an exemption determination.

9) Is the PGx results enough to change the administration? Are there any clinical guidelines that propose the use of PGx results to modify it? The authors state that this is not a study in human subjects after all so the clinical documentation should be enough for this case. If not and there are not clinical guidelines then this pilot study is a study in human subjects.  

We have provided information in section 2.5 (lines 151-158) and elaborated it how psychotropic guidance was used per the Food & Drug Administration (FDA), CPIC, or Dutch Pharmacogenetics Working Group (DPWG) [references 22-24]. In addition, we have added Supplementary Table S2, to provide more details on individual cases of patients who underwent PGx testing and received guidance.

10) 163-165. It is the essential part of the study, but no additional information is given. Which drugs, what interventions what actual clinical decisions were given based on PGx data.

We have provided an additional supplemental table (Table S1) with the medications that were assessed as part of our pilot. Additional language was added to line 151-158 describing how PGx data was used to make clinical decisions.

Please see Supplementary Table S1 and S2. This information is also part of Section 3.4 -lines 222-231.

11) Would expect at least a comparison of the clinical guidelines and the data from this study, how the utilization of PGx information improved the clinical decisions. An example, there are differences in doses since the study was referring to children and some adults. We do not see any discussion about those details that could further add in this study. We only see a brief discussion between this study and other studies with other disorders (lines 203-219).

Please see Supplementary Table S2 and section 3.4, how medications were changed after PGx test results-looking at metabolizer status of CYP2C19 and CYP2D6.

12) Too many abbreviations without description e.g., SSRIs etc.

Thank you pointing this out. We went through the manuscript and ensured all abbreviations were described prior to being used.

Round 2

Reviewer 1 Report

In this retrospective study, the multidisciplinary team found a high level of effectiveness for multidisciplinary/interdisciplinary consulting augmented with PGx testing in guiding medication therapy. 

The authors addressed all the queries and the manuscript was improved.

Author Response

Thank you for reviewing our article and providing helpful comments.  Our manuscript benefited with your suggestions in sections of Introduction, Results and Discussion.

Reviewer 2 Report

The retrospective study only intensifies something that has already been consolidated in the literature so far, there is no innovation for the theme.

Author Response

Thank you for reviewing our article and providing helpful comments.

Review of the literature shows there are only a few studies in the area of oncology where a multidisciplinary team assisted with PGx/genomic information and it has benefited the patient outcome. But in NDDs and NBDs, there are no well planned studies reported so far, except a few which report on multi specialities including Psychiatry. An example is Caraballo et al 2017 Genet Med 2017;19:421–9. and Brown et al Clin. Transl. Sci (2021) 14, 412–421.

To the best of our knowledge this is the first Retrospective Chart Review study which ilustrates the need for multidisciplinary pharmacogenomic consultation in NDDs/NBDs.

Reviewer 3 Report

The authors presented an updated version of their manuscript and tried to address the comments made from the reviewers. 

My biggest concern is the handling of samples but the authors seem to provide a comment that they obtained an informed consent from parents/guardians. The other serious issue is the phenoconversion which in this manuscript actually are cases of drug-drug interactions that the authors do not present or describe. Section 3.2 in Lines 196-205 I would expect the authors to present these drugs administrations and combinations. Moreover I would expect how these DDIs were avoided based on PGx data. On the other hand, is it clinical proper for adjunt therapies that may lead to DDIs? Patient 26 in supplementary may show pharmacodynamic synergism/antagonism cases (delirium, hypotensive effects, sedation, bradycardia). Patient 7 in risk for serotonin syndrome. 

"Phenoconversion is a phenomenon that converts genotypic extensive metabolizers (EMs) into phenotypic poor metabolizers (PMs) of drugs, thereby modifying their clinical response to that of genotypic PMs.(DOI: 10.1111/bcp.12441, ref 34)." Is this the case here? Was there actual change in the metabolic activity from the co-administration? I believe the authors considered it as de facto information although it is more complex in clinical practice. 
Even the references the authors use state that much. For example, ref 36 "Bupropion is therefore a potent inhibitor of CYP2D6 activity, and care should be exercised when initiating or discontinuing bupropion use in patients taking drugs metabolized by CYP2D6." On the other part some cases of phenocovnversion can be easily addressed with changes in dosing schemes (morning-evening) or if the drug is also substrate to other enzymes such as CYP3A4, the interaction although predicted not observed in clinical level or its severity is not so significant. But we do not see many details here regarding the medications, their ADRs, issues raised or how addressed for the first time etc. For example, the PGx profile would assist in these cases to avoid clinical significant DDIs that otherwise would be moderate (for EM), e.g., in the case of IM for example because the metabolic activity would drop in the presence of a CYP inhibitor. But apart of that would assist in dosing schedule (they are children after all) etc. etc. But we do not see many details. If a clinical pharmacist was included in the team would explain it better and why there is a strong need to present the cases. 

Ofcourse the work is interesting and the simple presentation for combining PGx information with clinical data would suffice for the manuscript to be accepted but I believe the authors tried to make it more complex instead to present the facts, after all is a pilot study. 

3.5 is still short in my opinion although the point of focus for this research. Table 2 in supplementary also is short. I would expect a detailed table and one placed within the manuscript. 

Author Response

Thank you for reviewing our article and providing helpful comments.  The manuscript benefited with your suggestions in the Introduction, Results and Discussion sections.

Comments and Suggestions for Authors

The authors presented an updated version of their manuscript and tried to address the comments made from the reviewers. 

My biggest concern is the handling of samples but the authors seem to provide a comment that they obtained an informed consent from parents/guardians. The other serious issue is the phenoconversion which in this manuscript actually are cases of drug-drug interactions that the authors do not present or describe.

On lines 164-166, we describe looking at this after the pilot to see the frequency. We used the publicly available calculator to determine if the patient had undergone phenoconversion.

Phenoconversion for CYP2D6 has been well established and described in pharmacogenetic guidelines, and most recently described in CPIC opioid guidelines. This is where it is described as a drug-drug-gene interaction.

We kindly disagree with this reviewer’s assessment of phenoconversion.

Section 3.2 in Lines 196-205 I would expect the authors to present these drugs administrations and combinations.

The line numbers have since shifted. We added the supplementary table that describes the gene-drug pairs. We have added an additional example to the supplementary Table 2 (now it is in the main text as Table 6) with an example of a patient 30 on Bupropion which is a strong CYP2D6 inhibitor. I hope we have addressed your concerns.

Moreover I would expect how these DDIs were avoided based on PGx data.

As mentioned in the methods section (line 151-162), phenoconversion was not assessed during the pilot, as the genetic testing laboratory reports did not include this information. We looked at it afterwards to help explain some of the challenges of using PGx and to inform the readers of considerations they should consider when implementing in a NDD/NBD clinic.

On the other hand, is it clinical proper for adjunt therapies that may lead to DDIs? Patient 26 in supplementary may show pharmacodynamic synergism/antagonism cases (delirium, hypotensive effects, sedation, bradycardia). Patient 7 in risk for serotonin syndrome. 

Patient 7 and patient 26 are on appropriate therapy for their diagnosis. It is part of standard of care to use multimodal therapy to treat NDD/NBD. Often patients will be on more than one therapy that works synergistically. This concept is often used in other conditions such as high blood pressure or diabetes.

"Phenoconversion is a phenomenon that converts genotypic extensive metabolizers (EMs) into phenotypic poor metabolizers (PMs) of drugs, thereby modifying their clinical response to that of genotypic PMs.(DOI: 10.1111/bcp.12441, ref 34)." Is this the case here? Was there actual change in the metabolic activity from the co-administration? I believe the authors considered it as de facto information although it is more complex in clinical practice. 

As mentioned previously our trial design was a pilot. We did not obtain samples to compare pharmacokinetic differences, nor did we collect pharmacodynamic outcomes. Being a pilot design about feasibility reporting on the above PK/PD is beyond the scope of this paper. Phenoconversion has been well published and well accepted in the pharmacogenomics community. We made sure in our paper to use conditional language and avoid definitive language (line 223-241). 

Even the references the authors use state that much. For example, ref 36 "Bupropion is therefore a potent inhibitor of CYP2D6 activity, and care should be exercised when initiating or discontinuing bupropion use in patients taking drugs metabolized by CYP2D6." On the other part some cases of phenocovnversion can be easily addressed with changes in dosing schemes (morning-evening) or if the drug is also substrate to other enzymes such as CYP3A4, the interaction although predicted not observed in clinical level or its severity is not so significant. But we do not see many details here regarding the medications, their ADRs, issues raised or how addressed for the first time etc.

As previously stated, it was not part of our pilot design, and collecting outcomes was not performed. This is a feasibility paper. In addition, suggesting altered dosing schemes would not address phenoconversion. For a medication to be considered eliminated, it takes approximately 5-half-lives. Thus, for example, fluoxetine has a half-live if chronically used of 4-6 days and would take approximately 20-30 days to be eliminated to no longer act as a CYP2D6 inhibitor. The concept of prolonged inhibition has been previously described (PMID: 33498694). This is why altering dosing schemes would not be appropriate. However, this concept would be appropriate if we were discussing alterations in the gastric pH by a proton pump inhibitor decreasing the absorption of another medication dependent on the acidic environment. 

PMID: 33498694

Deodhar M., Rihani S.B.A., Darakjian L., Turgeon J., Michaud V. Assessing the mechanism of fluoxetine-mediated CYP2D6 inhibition. Pharmaceutics. 2021;13:148.

For example, the PGx profile would assist in these cases to avoid clinical significant DDIs that otherwise would be moderate (for EM), e.g., in the case of IM for example because the metabolic activity would drop in the presence of a CYP inhibitor. But apart of that would assist in dosing schedule (they are children after all) etc. etc. But we do not see many details. If a clinical pharmacist was included in the team would explain it better and why there is a strong need to present the cases.

We are not certain what is being asked.  To help address these concerns we added an example of a drug inhibitor  (Bupropion  Table 6).

Ofcourse the work is interesting and the simple presentation for combining PGx information with clinical data would suffice for the manuscript to be accepted but I believe the authors tried to make it more complex instead to present the facts, after all is a pilot study. 

3.5 is still short in my opinion although the point of focus for this research. Table 2 in supplementary also is short. I would expect a detailed table and one placed within the manuscript. 

We appreciate your comments. Because this is a retrospective chart review study, we did not want to list too many of our patients in the Table to prevent conflicts with our IRB.  We chose an example of each type of possible PGx medication change classification for the table. It is unclear what else the reviewer is asking to be added to our supplementary table 2. As it is a chart review, we are limited on what we are allowed to report on. We will move the supplementary Table 2 into the manuscript (now Table 6).